# Omega-3 Fatty Acids for Depression in the Elderly and Patients with Dementia: A Systematic Review and Meta-Analysis

**DOI:** 10.3390/healthcare12050536

**Published:** 2024-02-23

**Authors:** Yen-Yun Chang, Berne Ting, Daniel Tzu-Li Chen, Wei-Ti Hsu, Song-Chow Lin, Chun-Yen Kuo, Ming-Fu Wang

**Affiliations:** 1Department of Food and Nutrition, Providence University, Taichung 433719, Taiwan; mfwang@pu.edu.tw; 2Ph.D. Program for Aging, College of Medicine, China Medical University, Taichung 404333, Taiwan; berne.ting@gmail.com; 3Department of Psychiatry and Mind-Body Interface Laboratory (MBI-Lab), China Medical University Hospital, Taichung 404327, Taiwan; rocket1025918@gmail.com; 4Graduate Institute of Biomedicine, College of Medicine, China Medical University, Taichung 404327, Taiwan; u108305203@cmu.edu.tw; 5College of Chinese Medicine, China Medical University, Taichung 404327, Taiwan; 6Department of Anesthesiology, China Medical University Hospital, Taichung 404327, Taiwan; 7Department of Medicine, Taipei Medical University, Taipei 11031, Taiwan; songchow@tmu.edu.tw; 8Ph.D. Program in Health and Social Welfare for Indigenous Peoples, Providence University, Taichung 433719, Taiwan; cykuo2@pu.edu.tw

**Keywords:** elderly, dementia, depression, fish oil, omega-3 fatty acid, docosahexaenoic acid, eicosapentaenoic acid

## Abstract

This study aimed to evaluate the efficacy of omega-3 fatty acid supplementation interventions in improving depression in patients with dementia. To achieve this objective, randomized controlled trials (RCTs) were identified from primary electronic databases, focusing on the relationship between omega-3 fatty acids and depression in patients with dementia. The primary outcome was the impact of omega-3 fatty acids on post-intervention depression in patients with dementia, with subgroup analyses conducted based on the type of intervention (docosahexaenoic acid (DHA) and eicosapentaenoic acid (EPA) combination), duration of intervention (3 months, 6 months, 12 months, ≥24 months), cognitive function (ranging from mild cognitive impairment (MCI) to severe dementia), and daily dosage (high, medium, low, applicable to both DHA and EPA). The study has been duly registered with PROSPERO (registration ID: CRD42023408744). A meta-analysis of five studies (n = 517) included in nine systematic reviews showed that omega-3 supplementation had a non-significant trend toward affecting depressive symptoms in patients with dementia (standardized mean difference (SMD): 0.147; 95% confidence interval (CI): −0.324 to 0.049; *p* = 0.141). Subgroup analyses revealed that DHA supplementation significantly reduced depressive symptoms (SMD: −0.247; *p* = 0.039). There was no significant effect for high (SMD: −0.169; 95% CI: −0.454 to 0.116; *p* = 0.246) or medium (SMD: −0.061; 95% CI: −0.228 to 0.105; *p* = 0.470) doses of EPA. However, low doses of EPA were significantly effective (SMD: −0.953; 95% CI: −1.534 to −0.373; *p* = 0.001), with notable improvements in patients with MCI (SMD: −0.934; *p* < 0.001). The study concludes that omega-3 fatty acids, particularly through DHA supplementation, may alleviate depressive symptoms in patients with MCI. Given the limited sample size, further long-term RCTs are recommended to better understand the efficacy and optimal management of omega-3 supplementation in this population using different dosages.

## 1. Introduction

Dementia and depression represent two prevalent disorders markedly impacting the elderly population [1]. Although distinct, these pathologies are somewhat intertwined, manifesting overlapping symptoms such as diminished memory and attention, social withdrawal, and diminished enjoyment in previously pleasurable activities. These overlapping characteristics pose challenges in diagnosis and treatment planning [2,3]. Dementia is a neurodegenerative condition hallmarked by a progressive decline in cognitive abilities, culminating in compromised daily living and independence. Various types of dementia have been identified, including Alzheimer’s disease (AD), vascular dementia, dementia with Lewy bodies, and frontotemporal dementia. The incidence of dementia escalates with advancing age, particularly in individuals over the age of 65 years [4,5]. Based on the World Health Organization’s estimates, dementia affects approximately 5% to 8% of the global elderly population [6]. On the other hand, depression is a widespread mental disorder characterized by persistent melancholy, diminished interest or pleasure in daily activities, and additional cognitive and physical symptoms such as fatigue, disrupted sleep, and impaired concentration [7]. The prevalence of depression is relatively higher amongst the elderly, estimated to afflict over 15% of the global geriatric population [8,9,10].

Treatment strategies for dementia and depression in the geriatric population are diverse, based on their diagnoses, symptom severity, and overall health status [8]. Although most dementia types are considered unreversible [11,12,13], symptom management can be achieved to improve patients’ quality of life [5]. Current treatment includes pharmacological therapy (e.g., first-line antidepressants like selective serotonin reuptake inhibitors (SSRIs) and serotonin and norepinephrine reuptake inhibitors (SNRIs) [14], non-pharmacological interventions (e.g., cognitive behavioral therapy (CBT), interpersonal therapy (IPT), electroconvulsive therapy (ECT)), and lifestyle modifications (e.g., encompassing regular physical exercise, adherence to nutritious dietary habits, ensuring sufficient sleep, and participation in social engagements [15]. However, the potential side effects of pharmacological and invasive therapy, such as gastrointestinal symptoms, dizziness, dry mouth, and anesthesia risk, should be taken into account [16,17,18].

By contrast, dietary supplements, such as omega-3 fatty acids, have been widely used to improve depressive symptoms and cognitive function clinically [19,20,21] for their effectiveness and safety [22]. Omega-3 fatty acids, particularly eicosapentaenoic acid (EPA) and docosahexaenoic acid (DHA) that are abundantly present in fish oil, are currently under thorough investigation for their potential efficacy in alleviating symptoms of depression and dementia. Several potential mechanisms have been identified to bolster brain health. For instance, encompassing neurotransmission, anti-inflammatory effects, neuroprotection, and improving neuroplasticity [23,24] may play pivotal roles in mood regulation and cognitive function [25]. However, while extensive research has been conducted on the effects of omega-3 fatty acids on depressive symptoms in the general population, the specific impact of omega-3 on elderly individuals and patients with dementia remains unclear and limited. Previous investigations of omega-3 fatty acids in patients with dementia mostly targeted general cognitive function [26,27]. However, the existing literature exhibits a significant gap in knowledge regarding the effects of different types of omega-3, such as DHA and EPA, and their dosages in this distinct demographic. This study aims to fill this void by providing a comprehensive systematic review and meta-analysis on the effects of omega-3 fatty acids on depressive symptoms, especially the core symptoms (depressed mood and loss of interest) of depression in the elderly and patients with dementia.

We hypothesized that omega-3 fatty acids are effective for depressive symptoms in patients with dementia, and it may be dosage-dependent. Our investigation therefore particularly focuses on the efficacy of different types and dosages of omega-3 supplements in this unique population, aiming to offer more detailed and specific evidence to inform clinical treatment guidelines.

## 2. Materials and Methods

### 2.1. Protocol and Registration

The meta-analysis conducted adhered to the PRISMA (Preferred Reporting Items for Systematic Reviews and Meta-Analyses) guidelines [28] (Appendix A). The study has been duly registered with PROSPERO, the International Prospective Register of Systematic Reviews (registration ID: CRD42023408744).

### 2.2. Search Strategy and Selection Criteria

The comprehensive search for relevant literature was conducted across four databases: PubMed, Embase, Web of Science, and the Cochrane Library, spanning their inception until January 2024. The search strategy deployed involved a Boolean search method, incorporating keywords such as omega-3 fatty acids, fish oil, docosahexaenoic acid (DHA), eicosapentaenoic acid (EPA), depression, Alzheimer’s disease, MCI, and memory loss. This approach enabled a comprehensive coverage of relevant literature across the domains of depression, dementia, Alzheimer’s disease, MCI, and related memory loss conditions, utilizing the intersection and union of these terms for an exhaustive search (Appendix A). The initial screening involved the removal of duplicate articles and those not specifically focusing on the elderly and patients with dementia. The search was guided using the PICO framework (Patient—depression in elderly patients with dementia; Intervention—omega-3 supplementation; Comparison—control groups receiving no intervention or non-pharmacological interventions; and outcome—assessment of depression in the elderly and patients with dementia). The titles and abstracts of the remaining articles were subsequently assessed. Inclusion criteria for this review encapsulated the following: (1) randomized controlled trials (RCTs) and non-randomized comparison studies (NRSs) published in peer-reviewed journals, providing sufficient statistical detail, such as means, standard deviations, and participant numbers for quantitative synthesis; (2) interventions that included fish oil components (EPA, DHA, or combination) in the intervention group and standard treatment, no treatment, or non-omega-3 interventions in the control group; (3) outcome assessments incorporating depression measures; (4) study populations that included participants aged 65 years and older; (5) participants diagnosed with dementia, MCI or participants with self-reported memory loss. Only studies that met these criteria were included in the qualitative synthesis. Exclusion criteria comprised the following: (1) articles that were reviews, medical protocols, conference proceedings, case reports, letters, editorials, pilot RCTs, and pilot studies; (2) omega-3 interventions combined with other therapies or as part of complementary or alternative therapies; (3) control groups that received any form of fish oil omega-3; (4) studies that lacked primary outcome analysis information; (5) studies without any comparison group or those not reporting any comparative results between groups; and (6) observational studies. Eventually, the full text of eligible articles was evaluated, leading to the final selection for meta-analysis.

### 2.3. Data Extraction

The synthesized data incorporated several variables, as delineated in Table 1. Each row corresponds to an individual study. The ‘General Characteristics’ section encapsulates details such as authors, publication year, the country where the study was conducted, study design, and the comparisons performed within the study. ‘Participant Characteristics’ conveys information regarding participant type, total number of subjects, their mean age, and the proportion of female participants in the study. The column titled ‘Intervention Components’ itemizes the type/genres of intervention, the dosage administered, and the duration of the intervention. The ‘Control Group’ column elucidates the nature of the control used in each study. Lastly, the ‘Outcome Measures’ column delineates the type of depression rating scales deployed in each study. The data extraction process was independently executed by two authors (Chang and Ting), and any inconsistencies were resolved via consensus-based discussions involving two other authors (Chen and Hsu).

### 2.4. Assessment of Methodological Quality

For assessing the methodological robustness of the included studies, we employed the Cochrane Collaboration’s Risk of Bias Tool for Randomized Trials (RoB 2, version 2, London, UK) [37]. This comprehensive tool thoroughly evaluates crucial aspects of research quality, such as the randomization process, adherence to intervention protocols, management of missing outcome data, precision in measuring outcomes, likelihood of selective reporting, and the general risk of bias present in the study. After assessing methodological quality, we completed the GRADE “Summary of Findings” and evaluated the certainty of the evidence [38]. This step systematically reviewed the evidence’s reliability, considering study design, execution, and consistency of results. The potential bias risk in the studies incorporated in this analysis was independently assessed by two authors (Ting and Hsu). Any differences in evaluations among the two authors were addressed through discussion with the third author (Chen).

### 2.5. Grouping of Omega-3 and Control Participants

The experimental cohort in this study was composed of participants who were administered omega-3 fatty acids derived from fish oil. The control group was divided into two distinct categories. The first category was designated as ‘placebo,’ comprising individuals who received a substance without any active treatment. The experimental cohort in this study was composed of participants who were administered omega-3 fatty acids containing EPA and/or DHA, to improve readability and reduce misunderstandings.

### 2.6. Primary Outcome: (Improvement in Depression in Patients with Dementia)

Meta-analysis results were assessed via standardized mean differences (SMDs) and 95% confidence intervals (CIs). The primary outcome was the alteration in depressive scores in patients with dementia after the omega-3 intervention, with data chosen from 3-month, 6-month, 12-month, and 24-month post-intervention evaluations if multiple measurements were available. The assessments and score-giving were conducted by an objective investigator. If the standard deviation (SD) or 95% CI were not utilized in the included studies, the analyses were estimated through a random effects model [39]. In studies featuring multiple arms, a single paired comparison was created and unrelated groups were excluded from the analysis. The effect size was deciphered following Cohen’s guidelines, with 0.2 denoting a small effect, 0.5 indicating a medium effect, and 0.8 implying a large effect. All *p*-values were two-tailed, with statistical significance set at 0.05. The dataset underwent processing via the Biostat Comprehensive Meta-analysis software, version 4. Data analysis was carried out by Ting and Hsu, with any arising disputes resolved by Chang and Chen.

### 2.7. Subgroup Outcomes

In the subgroup analysis, the data were segregated and evaluated based on several distinctive factors: (1) Intervention type: four groups were identified, which were DHA + other supplements (OS), DHA, DHA + EPA, and EPA. (2) Intervention duration: The follow-up durations post-intervention were grouped into four subcategories: 3 months, 6 months, 12 months, and ≥24 months. (3) Cognitive impairment severity level: the three categories were MCI, moderate, and severe. (4) Dosage of DHA and EPA: subgroups were formed based on the daily dosage of DHA and EPA provided to the participants, classified into ‘High’, ‘Moderate’, and ‘Low’ dose categories as defined below: DHA, high dose: 1.55 g and above, moderate dose: 0.6–0.7 g, low dose: 0.35 g and below; EPA, high dose: 1.6 g and above, moderate dose: 0.6–0.8 g, low dose: 0.4 g and below. (5) Rating scale: the final analysis considered the type of depression scale used to measure the outcomes. It was divided into two categories: those using the Geriatric Depression Scale (GDS) and those using other scales (non-GDS). Each of these subgroup analyses aimed to reveal any potential nuances or relationships that might otherwise be overlooked in a larger population. The intention was to uncover further insights into the intricate interplay between these factors, their combined effect on depression in patients with dementia, and the potential therapeutic efficacy of omega-3 supplementation.

### 2.8. Publication Bias

We assessed the presence of publication bias following the recommendations outlined in the Cochrane Handbook for Systematic Reviews of Interventions [40]. To visualize potential bias, we generated a funnel plot using Comprehensive Meta-Analysis software, version 4 (Biostat, Englewood, NJ, USA), focusing on comparisons that included the control group. Furthermore, Egger’s regression test was performed to identify any noteworthy publication bias.

## 3. Results

### 3.1. Identification of Eligible Studies

During the identification stage, our database search yielded 1100 studies. After the removal of 977 duplicate entries, 123 unique studies remained. These were subjected to the screening phase, where the titles and abstracts of these articles were closely reviewed. This led to the exclusion of 92 studies that were not relevant to our research objectives. Subsequently, 31 full-text articles were chosen for an in-depth eligibility assessment. At this point, we excluded 17 studies since our analysis focused primarily on RCTs (Appendix A). As a result, we included nine studies in our qualitative synthesis during the inclusion phase. After rigorous evaluation, only five of these studies met all the stipulated criteria, and hence they were included in the meta-analysis (Figure 1).

### 3.2. Study Characteristics and Patient Population

We analyzed a total of nine studies encompassing a range of participant types, interventions, and outcome measures [21,29,30,31,32,33,34,35,36]. In terms of general characteristics, the studies span 2008 to 2022. The selected studies were conducted in diverse geographic regions, including Sweden, Netherlands, Japan, Taiwan, France, Italy, and Australia. All studies were RCTs, while some were designed as multi-arm. Interventions varied across studies and included DHA, EPA, combined DHA+EPA, DHA with a previously defined OS (including vitamin E, soy phospholipids such as phosphatidylinositol, phosphatidylcholine, and phosphatidylethanolamine, tryptophan, and melatonin), and placebo. Comparisons were made to various control groups, predominantly placebos. The participant characteristics were also diverse, ranging from patients’ severity level with MCI, to moderate-to-severe cognitive impairment. The number of participants in each study varied from as few as 25 to as many as 1680. The participants’ mean age ranged from 69 to 88 years, and the percentage of female participants varied between 18% and 86.67%. The intervention components consisted of different types of omega-3 fatty acids, including DHA, EPA, and their combination in different doses. The duration of the interventions ranged from 3 months to 36 months. The outcome measurement varied among the studies, including depression rating scales such as the Montgomery Asberg Depression Scale (MADRS), Zung Self-Rating Depression Scale (SDS), and Geriatric Depression Scale (GDS) (Table 1).

### 3.3. Quality Assessment of Included Articles

In our evaluation of the methodological quality of the nine studies included, the following observations were made. Randomization process: all studies (100%, 9/9) were evaluated to have a low risk of bias. Intervention adherence: concerns were noted in one study (11.1%, 1/9), while the rest (88.9%, 8/9) demonstrated a low risk of bias. Missing outcome data: similar to intervention adherence, one study (11.1%, 1/9) raised some concerns, but the majority (88.9%, 8/9) were found to have a low risk. Outcome measurement: consistently, all studies (100%, 9/9) were rated as having a low risk of bias in measuring outcomes. Selective reporting: likewise, all studies (100%, 9/9) were assessed as showing a low risk of bias. Overall risk of bias (RoB): unanimously, all studies (100%, 9/9) were rated as having a low overall risk of bias (Appendix A). In summary, the studies included in this analysis showed a strong methodological quality, with all domains predominantly indicating a low risk of bias. A particular area of concern was intervention adherence, where a significant majority of studies indicated some concerns. However, despite these concerns, the overall risk of bias across the studies remained low. This robust methodological quality suggests a high level of reliability in the outcomes reported by these studies. Comprehensive information on the risk of bias evaluation is presented in Appendix A. Evidence quality grades for high and moderate studies indicate moderate, which means that these results are based on some good-quality studies but may include some biases, such as design flaws or consistency issues in outcomes, which could affect the accuracy and reliability of the results (see Appendix A).

### 3.4. Primary Outcome: Effect of Omega-3 Fatty Acids for Depression in Elderly Patients with Dementia

Figure 2 presents the forest plot of the meta-analysis evaluating the impact of omega-3 supplementation on depressive symptoms in elderly patients with dementia; among the nine studies synthesis reviewed, which together included a total of 3932 participants, data from five studies encompassing 517 participants were synthesized for our primary outcome of meta-analysis, showing an insignificant result. The standardized mean difference (SMD) in depressive symptoms between the omega-3 group and the control group was 0.147. This indicates a slight, yet not statistically significant, improvement in depressive symptoms for those receiving omega-3 supplementation. The 95% confidence interval ranged from −0.324 to 0.049, suggesting that the true effect might range from a modest worsening to a modest improvement. It should be noted that the *p*-value of 0.141 suggests that the observed effect size is not statistically significant at the conventional 0.05 threshold. Regarding heterogeneity, the I^2^ statistic, which measures the percentage of total variation across studies that is due to heterogeneity rather than chance, was 0. This suggests that there was no observed heterogeneity among the studies included in this meta-analysis. The *p*-value for I^2^ was 0.529, further indicating that the lack of observed heterogeneity was not statistically significant. Egger’s regression test produced a *p*-value of 0.10. Furthermore, the funnel plot analysis indicated that the effect sizes were distributed within the diagonal lines, implying an equitable distribution and reducing the likelihood of a significant publication bias (Appendix A).

### 3.5. Subgroup Analysis

In a detailed subgroup analysis examining the impact of different intervention types, derived from the data of 16 arms included within the five meta-analyzed studies, we observed distinct outcomes. For the EPA, the effect size (SMD) was −0.169 (*p* = 0.246). The DHA showed an SMD of −0.247, significant at *p* = 0.039. The combination of DHA and EPA resulted in an SMD of −0.061, yielding a non-significant *p*-value of 0.470. Lastly, the DHA+OS demonstrated a pronounced effect size of −0.815, narrowly missing significance with *p* = 0.052. Analyzing the data based on intervention duration, for a 3-month duration (3 m), the effect size (SMD) was −0.815 (*p* = 0.052). For the 6-month (6 m) period, the SMD was −0.177 (*p* = 0.146). Over a 12-month (12 m) span, the SMD was −0.147 with a *p*-value of 0.120. Lastly, for the 24-month period or a longer (≥24 m) duration, the SMD stood at −0.138, resulting in a *p*-value of 0.331. Upon stratifying the data by cognition, for MCI, the effect size (SMD) was −0.934, significant at *p* < 0.001. In the moderate-dementia subgroup, the SMD was −0.073 (*p* = 0.265). For those diagnosed with severe dementia, the SMD was −0.295, nearing a statistically significant *p*-value of 0.076. When segmenting the data by daily DHA dosage, for a high dosage, the effect size (SMD) was −0.226, yielding a *p*-value of 0.085. In the moderate dosage category, the SMD was −0.172 (*p* = 0.203). Lastly, the low dosage subgroup indicated an SMD of −0.156, resulting in a *p*-value of 0.267.

Breaking down the data by daily EPA dosage, for the high dosage subgroup, the effect size (SMD) was −0.169, associated with a *p*-value of 0.246. In the moderate dosage category, the SMD was −0.061 (*p* = 0.470). For the low dosage subgroup, the SMD was notably −0.953, significant at *p* = 0.001. Classifying the data based on the rating scale used, for studies utilizing the GDS (Geriatric Depression Scale), the effect size (SMD) was −0.172, which was statistically significant with a *p*-value of 0.031. For the non-GDS category, the SMD was −0.126, resulting in a non-significant *p*-value of 0.163. The studies reviewed were diverse in design, participant profiles, and omega-3 dosages, which could affect the outcomes, especially regarding the impact on different depression levels. Hence, this variability is important when evaluating our findings. Detailed results are presented in Table 2.

## 4. Discussion

In this meta-analysis, we conducted a comprehensive evaluation of the effects of omega-3 fatty acid supplementation on depressive symptoms in the elderly population, including those diagnosed with dementia. The findings of this study provide clinicians with information on the potential benefits of specific omega-3 fatty acid compounds for geriatric depressive symptoms. While the overall results indicate no widespread positive impact of omega-3 fatty acids on depressive symptoms in the elderly, our data analysis unveiled some promising subtle differences. Specifically, our subgroup analysis highlighted a significant positive effect of DHA in alleviating symptoms (*p* = 0.039). Moreover, a notable improvement was observed in individuals with mild cognitive impairment (MCI) (*p* < 0.001). Previous studies have generally suggested that omega-3 fatty acids possess antidepressant properties [22,41]. However, their efficacy may vary when administered to the elderly [42]. Regrettably, studies focused on this particular age group remain scarce. Although EPA is widely recognized as the most potent antidepressant compound within the range of omega-3 fatty acids, there is still some controversy over the role of DHA [43,44]. Nevertheless, our study underscores the significance of omega-3 fatty acids, suggesting different mechanisms of action across various populations.

Our second finding was evidence indicating a positive effect of omega-3 supplementation on patients diagnosed with MCI, but not in those with moderate-to-severe cognitive impairments. This interesting finding may indicate that the effectiveness provided by omega-3 fatty acids may be highly associated with the disease severity. In contrast, this beneficial effect was not apparent in those with moderate-to-severe cognitive impairments [22,45,46]. Dementias, such as Alzheimer’s disease, represent complex neurodegenerative disorders of which the etiology is not fully understood. Current knowledge emphasizes its association with various biological phenomena, including the appearance of neurofibrillary tangles, accumulation of amyloid plaques, neurotoxicity leading to cell death, inflammatory cascades, and oxidative stress [29,30]. Studies suggest that omega-3 fatty acids, including DHA and EPA, are renowned for their cardiovascular benefits [47]. They have demonstrated positive effects on cognitive function in animal models and some human studies [48,49]. Omega-3 fatty acids participate in neuroprotection, and anti-inflammatory and antioxidant activities, all of which are beneficial for managing neurodegenerative diseases like Alzheimer’s disease [1]. Moreover, due to wide investigations, guidelines for omega-3 fatty acids used as antidepressants have been published [20]. However, our meta-analysis found that while omega-3 supplementation showed significant therapeutic benefits for those with MCI, those with moderate-to-severe cognitive impairment did not experience the same degree of alleviation in depressive symptoms. This could be attributed to several factors. First is the disease stage: AD is a progressively developing condition, and by its later stages, a substantial neuronal loss occurs. At this juncture, even with omega-3 intake, it may be challenging to reverse the already-manifested neuronal damage [50]. Second is bioavailability: some studies indicate that the metabolism of omega-3 might be altered in AD patients, meaning that even with omega-3 supplementation, effective concentrations might not be achieved in the brain [51]. Third is genetic factors: certain genetic mutations might impact the efficacy of omega-3. For instance, individuals carrying the APOE4 allele may not respond as well to omega-3 as others [21,52]. In conclusion, while omega-3 may be beneficial for certain populations, it does not necessarily benefit everyone. This may be the result of the complex interplay of biological factors. It also underscores the importance of personalized medicine, implying that the best therapeutic strategies should be tailored based on each individual’s circumstances.

Another of our findings focused on the dosage of omega-3 fatty acids. Our data suggest that a lower dosage of EPA has a more pronounced effect compared to a higher dosage. This may imply the existence of a saturation threshold, beyond which increasing the dosage may not provide additional benefits. Based on lots of previous studies, the antidepressant effect of EPA is much higher than DHA due to its anti-neuroinflammation effect [53]; by contrast, DHA may be more effective on structural-related brain dysfunction or degeneration, such as AD and MCI since it is an essential component of brain structure synthesis [54]. Such a phenomenon might be due to the body’s limited capability to process excessive fatty acids from omega-3 supplementation [55,56]. However, reconstruction of the brain structure may be much more difficult than reducing the inflammation status, and that may explain our interesting findings. While the effects of DHA were not statistically significant at different dosages, its overall effect was significant in alleviating depressive symptoms in patients with dementia. The insignificant results from the DHA subgroup may indicate that the reconstructed effect of DHA may not be dosage dependent.

The absorption of DHA and EPA in the elderly is a multifaceted process influenced by various factors, including dietary composition, overall health status, and individual metabolic rates. Studies focusing on the elderly population suggest that several factors may hinder the absorption and subsequent utilization of DHA and EPA. Firstly, aging is often accompanied by a decline in digestive system efficiency, potentially affecting the absorption and utilization of various nutrients, including DHA and EPA [57,58]. Current literature suggests that, with age, pancreatic fibrosis and atrophy are evident, leading to reduced exocrine function, even in the elderly without gastrointestinal pathology. Pancreatic exocrine insufficiency (PEI) is associated with an indirect malabsorption of fats and malnutrition, estimated to affect about 5–10% of healthy individuals over 70 [59]. Secondly, aging is associated with metabolic alterations, including fat metabolism. These changes might influence how the elderly utilize omega-3 [60]. Prior research has established the relationship between age-related muscle atrophy and an increase in visceral fat, accompanied by slight fat infiltration into muscle fibers, resulting in a condition known as myosteatosis. This situation subsequently affects the body’s lipid storage and metabolism [61]. Notably, the impact of physical activity on postprandial lipid concentrations is influenced by many variables, including the frequency, mode, and duration of exercise, meal composition, energy intake, and timing of physical activity. It is speculated that one of the primary mechanisms influencing the reduction in postprandial lipid concentrations induced by physical activity is the enhancement of lipoprotein lipase (LPL) activity. After exercise, fatty acid metabolism is accelerated, so sustained exercise training might reduce lipid concentrations post high-fat meal intake [62]. Moreover, this emphasizes the need to consider multiple factors when prescribing omega-3 supplementation, especially in the elderly, and reinforces the value of individualized dosing. Chronic conditions: As numerous studies have indicated, a prevalent incidence of chronic diseases such as cardiovascular disease and diabetes among the elderly can impact lipid metabolism, thereby affecting the absorption and utilization of DHA and EPA [63]. Lastly, many pharmaceuticals routinely prescribed for the geriatric population may interfere with the absorption or metabolic processing of certain nutrients. Specifically, some lipid-lowering agents have been implicated in altering the uptake of lipids, including DHA and EPA [64]. In summary, while the health benefits of omega-3 supplementation are evident in the elderly and those with dementia, excessive intake may not provide additional advantages and could lead to adverse outcomes. It is crucial to calibrate omega-3 dosages carefully, considering individual needs. The interaction between statin medications and the metabolism of omega-3 fatty acids is also noteworthy, with the impact varying depending on the statin type [65].

## 5. Limitations

Nevertheless, the inherent limitations of our study must be acknowledged. The low heterogeneity observed, while indicative of the reliability of our findings, may limit the exploration of other external factors that could potentially affect the outcomes. Moreover, given that the majority of our subgroup analyses are based on a limited number of studies, these findings necessitate further empirical confirmation.

## 6. Conclusions

In essence, our meta-analysis supports the selective use of omega-3 fatty acids, notably DHA, in the amelioration of depressive symptoms in older adults with MCI. It calls for individualized treatment modalities that respect the unique metabolic and health status of the elderly. The preliminary nature of our insights beckons further research, such as long-term follow-up RCTs with different dosage groups to cement these findings and refine omega-3-based supplementation strategies for this delicate demographic. Moreover, the implications of these findings extend beyond the clinical sphere, suggesting a recalibration of how we perceive and administer omega-3 supplements within personalized medicine paradigms for the aging population.

## Figures and Tables

**Figure 1 healthcare-12-00536-f001:**
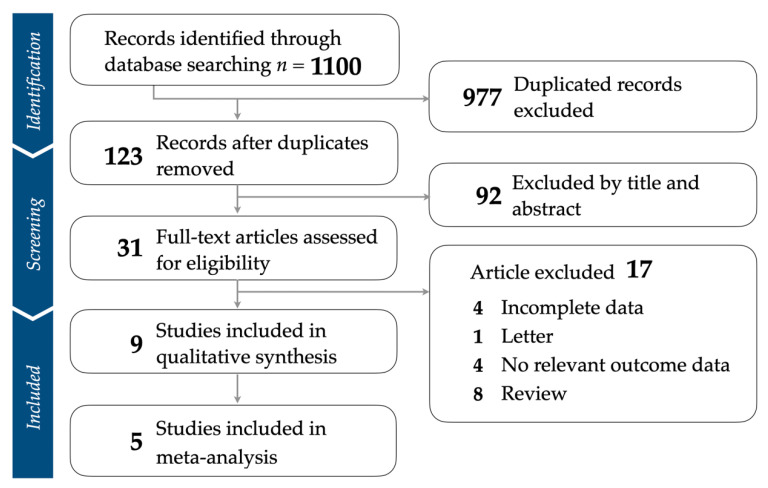
Flowchart illustrating the strategy for study selection and the application of inclusion and exclusion criteria.

**Figure 2 healthcare-12-00536-f002:**
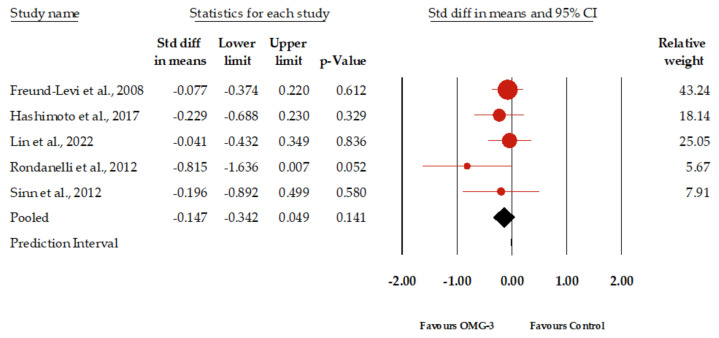
Forest plot of the graphical representation of the impact of omega-3 supplementation on depressive symptoms in patients with dementia [21,29,32,34,35].

**Table 1 healthcare-12-00536-t001:** Overview of the study characteristics.

Authors and Year	Country	Dose and Method	Sample	Treatment Duration	Results	Outcome	Type/Genres
Freund-Levi et al., 2008 [29]	Sweden	Dose: 1.7 g DHA and 0.6 g EPA daily. Method: Randomized, double-blind, placebo-controlled. Patients received either omega-3 supplements or placebo.	Size: 204 patients enrolled, 174 completed the study. Characteristics: Patients with Alzheimer’s disease, living in their own homes, stable dose of acetylcholine esterase inhibitors. Mean age: 74 years. Gender distribution: 51% female.	12 months (6 months randomized to omega-3 or placebo, followed by 6 months of omega-3 for all).	No significant overall treatment effects on neuropsychiatric symptoms, daily living activities, or caregiver’s burden. Some significant effects observed in specific domains (NPI agitation domain in APOEv4 carriers, MADRS scores in non-APOEv4 carriers).	MADRS	
Freund-Levi et al., 2014 [30] *	Sweden	Dose: 430 mg DHA and 150 mg EPA per capsule, total of four capsules daily. Additional Components: 4 mg vitamin E per capsule. Placebo: Corn oil with 0.6 g linoleic acid per capsule. Method: Randomized, double-blind, placebo-controlled. Patients received either omega-3 supplements or placebo oil.	Size: 41 patients enrolled, 37 completed the study. 17 omega-3 fatty acid (ω-3 FA) group. 20 placebo group. Characteristics: Patients diagnosed with Alzheimer’s disease according to the DSM-IV criteria, having an MMSE score between 15 and 30. Mean age: 70 years. Gender distribution: 40.5% female.	6 months.	No significant difference in treatment effect between supplemented and non-supplemented patients.	-	
Giudici et al., 2020 [31] *	Netherlands	Dose: 400 mg DHA and ≤112.5 mg EPA per capsule, two capsules daily. Method: Randomized assignment into one of four groups: 1. Multidomain intervention plus omega-3 supplementation; 2. Multidomain intervention plus placebo; 3. Omega-3 supplementation only; 4. Placebo. Multidomain intervention components: Cognitive stimulation, physical activity, and nutritional counseling.	Size: 1445 participants. Characteristics: Participants must have self-reported memory loss and inability to carry out an instrumental activity of daily living. Mean age: 75.3 years. Gender distribution: 64.2% female.	36 months.	No significant between-group differences were observed in the decrease in intrinsic capacity (IC) Z-score among all groups after 3 years when comparing each intervention group with participants taking placebo. Additionally, no long-term effects were noted in the omega-3 supplementation groups compared to placebo.	-	
Hashimoto et al., 2017 [32]	Japan	Dose: 860 mg DHA and 203 mg EPA daily for the intervention group; 53 mg DHA and 15 mg EPA daily for the control group. Method: Double-blind study. Participants received daily cooked meals including fish sausages, with differing DHA and EPA contents. Participants were blinded to the food products.	Size: 75 participants enrolled, 66 completed the study. Characteristics: Patients with Alzheimer’s disease, elderly individuals from care facilities and a nursing home in Shimane prefecture, Japan, including 10 men and 65 women. Mean age: 88.5 years. Gender distribution: 86.67% female.	12 months.	Significant increases in the levels of EPA and DHA, as well as in the DHA/AA and EPA/AA ratios, were observed in the active group compared to the control group after 12 months. Furthermore, certain aspects of cognitive functions, measured by MMSE and HDS-R scores, demonstrated significant improvements, suggesting benefits against age-related cognitive decline.	SDS	
Lin et al., 2022 [21]	Taiwan	Dose: Pure EPA group: 1.6 gm daily; Pure DHA group: 0.7 gm daily; Combination group: 0.8 gm EPA and 0.35 gm DHA daily. Method: Participants were divided into three groups, each receiving two 0.5 gm capsules twice a day. The placebo group received soybean oil capsules.	Size: 163 participants. Characteristics: Condition distribution: 93 with MCI, 70 with mild to moderate Alzheimer’s disease. Source: Mainly elderly individuals from veteran retirement centers. Mean age: 77.9 years. Gender distribution: 33.75% female.	24 months.	No significant differences were observed among treatment groups and placebo regarding cognitive, functional ability, and mood status outcomes after 24 months. This suggests that n-3 PUFA supplementation might not have significant effects on the disease progression rate in MCI and AD patients.	GDS	
Maltais et al., 2019 [33] *	France	Dose: Two soft capsules daily, each containing 400 mg DHA and up to 112.5 mg EPA. Method: Participants were assigned to one of four groups for a duration of 3 years: 1. Multidomain intervention plus omega-3; 2. Omega-3 only; 3. Multidomain intervention only; 4. Placebo. Multidomain intervention components: Nutritional and physical activity counseling, and cognitive training.	Size: 1680 participants. Characteristics: Participants must have self-reported memory loss, and slow gait speed. Demographics: Community-dwelling men and women aged 70 years or older. Mean age: 75.2 years. Gender distribution: 64.73% female.	36 months.	No significant effect of any intervention (multidomain, omega-3, or their combination) was found on the progression of depressive symptoms. The incidences of developing minimum clinically meaningful, moderate, and severe depressive symptoms were not significantly different across the four groups.	-	
Rondanelli et al., 2012 [34]	Italy	Dose: Two capsules daily, each containing DHA 720 mg, EPA 286 mg, vitamin E 16 mg, soy phospholipids 160 mg, tryptophan 95 mg, and melatonin 5 mg, taken 1 h before bedtime. Method: Randomized, double-blind, placebo-controlled. Participants received either the described capsules or placebo capsules containing non-fish oils without omega-3 or omega-6 fatty acids.	Size: 25 participants: 11 supplement group; 14 placebo group. Characteristics: Participants with MCI. Mean age: 86 years. Gender distribution: 80.20% female.	12 weeks.	Not statistically significant, the trend of improvement in depressive symptoms as evaluated with GDS was observed in the supplement group after 12 weeks of treatment with the dietary supplement.	GDS	
Sinn et al., 2012 [35]	Australia	Dose: EPA-rich fish oil group: 1.67 g EPA and 0.16 g DHA; DHA-rich fish oil group: 1.55 g DHA and 0.40 g EPA; Control group: Safflower oil with 2.72 g linoleic acid (LA). Method: Randomized allocation into one of three groups.	Size: 50 volunteers recruited, 40 completed assessments. Characteristics: Participants must have self-reported memory loss. Mean age: 74.03 years. Gender distribution: 33% female.	6 months.	Significant improvement in depression (GDS) scores was observed in the EPA and DHA groups compared with the control group after 6 months. However, no significant treatment effects were found on the physical or mental quality of life outcomes. In the DHA treatment group, initial letter fluency scores significantly improved compared with the control group.	GDS	
van de Rest et al., 2009 [36] *	Netherlands	Dose: High dose group: 1800 mg per day of EPA-DHA; Low dose group: 400 mg per day of EPA-DHA; Placebo group. Method: Randomized, double-blind, placebo-controlled. Participants received either 1800 mg, 400 mg of EPA-DHA, or a placebo.	Size: 302 participants. Characteristics: Participants were aged 65 years and older. A score of more than 21 on the MMSE. Mean age: 69.8 years. Gender distribution: 55% male.	26 weeks.	No significant differences were observed in the quality of life scores among the three groups after 13 and 26 weeks of intervention.	-	

Abbreviations: DHA (docosahexaenoic acid), EPA (eicosapentaenoic acid), LA (linoleic acid), MCI (mild cognitive impairment), SDS (Zung Self-Rating Depression Scale), GDS (Geriatric Depression Scale), * (Review only).

**Table 2 healthcare-12-00536-t002:** Subgroup analysis examining the effect of omega-3 on depression in patients with dementia.

Subgroup	*k*	Effect Size (SMD)	95% Confidence Interval	*p*
Intervention type				
EPA	4	−0.169	−0.454 to 0.116	0.246
DHA	6	−0.247	−0.482 to −0.013	0.039
DHA+EPA	5	−0.061	−0.228 to 0.105	0.470
DHA+OS	1	−0.815	−1.636 to 0.007	0.052
Length				
3 m	1	−0.815	−1.636 to 0.007	0.052
6 m	7	−0.177	−0.415 to 0.062	0.146
12 m	5	−0.147	−0.334 to 0.039	0.120
≥24 m	3	−0.138	−0.417 to 0.140	0.331
Cognition				
MCI	3	−0.934	−1.412 to −0.456	< 0.001
Moderate	11	−0.073	−0.200 to 0.055	0.265
Severe	2	−0.295	−0.620 to 0.031	0.076
DHA/Dosage (Day)				
High	5	−0.226	−0.484 to 0.031	0.085
Moderate	4	−0.172	−0.436 to 0.093	0.203
Low	4	−0.156	−0.432 to 0.119	0.267
EPA/Dosage (Day)				
High	4	−0.169	−0.454 to 0.116	0.246
Moderate	5	−0.061	−0.228 to 0.105	0.470
Low	2	−0.953	−1.534 to −0.373	0.001
Rating scale				
GDS	12	−0.172	−0.329 to −0.016	0.031
non-GDS	4	−0.126	−0.302 to 0.051	0.163

Note: SMD: standardized mean difference; k: number of studies; DHA: docosahexaenoic acid; EPA: eicosapentaenoic acid; OS: other supplements; MCI: mild cognitive impairment; GDS: Geriatric Depression Scale.

## Data Availability

The study’s data can be found in the article and its Appendix A.

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
