# Peer review of "Omega-3 Fatty Acids for Depression in the Elderly and Patients with Dementia: A Systematic Review and Meta-Analysis"

_healthcare, 2024, doi:10.3390/healthcare12050536_

Round 1

Reviewer 1 Report

Comments and Suggestions for Authors

Introduction:

1.     Clarify the specific types of depressive symptoms targeted in the study.

2.     Provide a concise overview of existing literature on omega-3 fatty acids and depression in the elderly.

3.     Clearly state the research questions or hypotheses guiding the study.

Methods:

4.     Specify the databases used for the search and the search strategy employed.

5.     Justify the choice of the Cochrane Collaboration tool for quality assessment.

Results:

6.     Provide additional information on the observational study that was excluded during the analysis.

Discussion:

7.     Expand on the potential reasons for the observed differences in efficacy between DHA and EPA.

8.     Discuss the implications of the findings in the context of the existing literature on omega-3 fatty acids and depression.

9.     Provide a more detailed explanation of the observed effects in individuals with Mild Cognitive Impairment (MCI).

10.  Consider discussing any potential biases that might have affected the study outcomes.

General:

11.  Consider providing a more detailed explanation of the rationale for the selected studies in the qualitative synthesis.

12.  Ensure consistency in terminology and abbreviations throughout the manuscript.

Reviewer 2 Report

Comments and Suggestions for Authors

The authors present a manuscript aimed to discuss the impact of omega-3  supplementation on patients with dementia, particularly concerning depressive symptoms, with a SR and Meta-analysis.

Despite they emphasize the fact that this topic has been assessed widely, they suggest the focused population (elderly) must be seen differently, in this sense, it could be useful to reinforce the need of this particular assessment in the introduction section.

Also, as proposed in the objective, table 1 must be enriched with enough information to respond to it, efficacy has not been included, kind of supplementation (oil, capsules or some other pharmacological form), as well as the main characteristics of the interventions. Additional information could be useful to understand the whole effects, as some factors could influence the final results; administration, kind of patients –if they were at some nursing home, shelter or home, using the same units to compare the dose, the base of the omega (if they are from oil fish or some other source, if the included studies report some adverse effects or inconvenience with the doses, and some other important information that could lead the author to understand and compare –if possible- the included studies.

On the other hand, none outcome has been reported in the main table (Table 1), just the used instrument.

Focusing in the meta-analysis, it is unclear if the administration of omega 3 could lead to an improvement, as seen in the forest plot, How do the authors explain that graphic.

Other aspect that arise during the interpretation of the results and discussion is the fact that the dose doesn’t improve the symptomatology (which has not been described before), this is quite shocking as it has been suggested that the effect could be greater in supplementations with higher doses (3 g/d) in some cases, contraire with the reported doses inhere, which could be considered as the recommendations for these group.

I suggest the authors to improve the presentation of the results in the SR section as well as the discussion in order to respond properly to the main objective of the study.

Reviewer 3 Report

Comments and Suggestions for Authors

This was a systematic review and meta-analysis that evaluated the impact of EPA and/or DHA supplementation on depressive symptoms in elderly adults with dementia. Nine articles were included in the qualitative analysis, and five articles were included in the quantitative meta-analysis. The results of the meta-analysis indicated no impact of omega-3 supplementation on depressive symptoms in patients with dementia. However, the subgroup analyses demonstrated that DHA supplementation alone and low doses of EPA may decrease depressive symptoms. Furthermore, the most pronounced improvements were in individuals with mild cognitive impairment. I have included a list of specific comments/suggestions. In general, the manuscript was well-written and would be of interest to readers. My major concern is the lack of assessments for publication bias (e.g., funnel plots) and quality of the evidence (e.g., GRADE assessment).

·       Line 31: Consider using people-first language (“patients with dementia” instead of “dementia patients”).

·       Line 90: “While” should not be capitalized.

·       Line 110: Why didn’t the search strategy include terms like EPA, DHA, MCI, etc.?

·       Line 163: Why didn’t you include omega-3 fatty acids derived from non-fish sources such as algae? Were any studies excluded from the systematic review and meta-analysis because of these criteria? Should this sentence be edited to say, “The experimental cohort in this study was composed of participants who were administered omega-3 fatty acids derived from EPA and DHA?

·       Line 222: Other supplements (OS) was already defined in line 186.

·       Line 222: Please give examples of other supplements that were used in conjunction with DHA in the included studies.

·       Line 224: Please give a few examples of the active interventions.

·       Line 188: Did you use ranges for the four subgroup categories for intervention duration? As written, it sounds like the 24-month subgroup only included studies that were 24 months in duration. Should this actually be the ≥24 months subgroup?

·       Did you assess publication bias?

·       Did you assess the quality of the evidence using the GRADE assessment tool?

·       Line 323: Please also note in this paragraph that only 2 studies were included in the severe cognitive impairment category. Therefore, there may not have been enough power to detect a difference in this population.

·       Line 355: I am not sure that the existence of a saturation threshold explains why the low-dose EPA subgroup had a more pronounced effect compared to the moderate and high dose EPA subgroups. If there was a saturation effect, then wouldn’t you see a similar effect in all 3 subgroups because they would all be reaching the level of saturation?

·       Lines 362-399: This is a very long paragraph. I suggest breaking it up into multiple paragraphs.

·       Lines 365, 366, 372, 386: I suggest reducing the number of colons used and instead using appropriate transitions. You could list the factors (reduced digestive efficiency, alterations in fat metabolism, chronic conditions, drug interactions) that may hinder absorption and utilization in line 365, and then describe them in more detail in the subsequent sentences.

·       Line 410: I suggest providing more specific examples of further research that should be conducted to address the gaps in the literature.

Round 2

Reviewer 3 Report

Comments and Suggestions for Authors

Thank you for taking the time to address prior comments. The quality of the manuscript has been improved. A few specific comments are outlined below.

Specific Comments to the Authors

·       Line 22, 193, 304, Table 2: The subgroup categories for the post-intervention duration still are not updated (e.g., ≥24 months). Please correct throughout the manuscript.

·       Line 89: Limited is misspelled.

·       Line 96: Please use people first language (patients with dementia).

·       The ROB 2 tool is useful for assessing the risk of bias of each individual study. Conversely, the GRADE assessment tool is the most widely used tool for grading the quality of the overall evidence. Risk of bias is just one of 5 domains that is assessed using the GRADE assessment. See section 14.2 of the Cochrane Handbook for Systematic Reviews of Interventions (Version 6.4; 2023) for a detailed explanation of why the GRADE assessment is important. Although I recognize that not every paper includes both assessments, I recommend including a GRADE assessment in addition to the ROB 2 assessment.

·       Line 349: Remove the period after MCI.
